# Evaluating the feasibility, fidelity, and preliminary effectiveness of a school-based intervention to improve the school participation and feelings of connectedness of elementary school students on the autism spectrum

Amy Hodges[1], Reinie Cordier[1,2]*, Annette Joosten[1,3], Helen Bourke-Taylor[1,4], Yu-Wei Chen[5]

1 School of Allied Health, Curtin University, Perth, WA, Australia, 2 Department of Social Work, Education and Community Wellbeing, Northumbria University, Newcastle, United Kingdom, 3 School of Allied Health, Australian Catholic University, Melbourne, VIC, Australia, 4 Department of Occupational Therapy, School of Primary and Allied Health Care, Monash University, Frankston, VIC, Australia, 5 Faculty of Medicine and Health, The University of Sydney, Sydney, NSW, Australia

* reinie.cordier@northumbria.ac.uk

## Abstract

*In My Shoes* is a peer supported, teacher-led, school-based intervention that aims to improve the school participation and connectedness of students on the autism spectrum. The aim of this study was to explore the feasibility, fidelity, and preliminary effectiveness of *In My Shoes* in mainstream elementary schools. Ten Grade 3 and 4 students on the autism spectrum and 200 of their typically developing peers across eight classrooms and six schools participated. The following aspects of feasibility were explored: recruitment capability and sample characteristics, data collection procedures and outcome measures, appropriateness, implementation, and practicality of the intervention. Fidelity was explored by evaluating the delivery of intervention components against set criteria. Preliminary effectiveness was investigated by evaluating changes in intervention outcomes pre-post intervention using a range of outcome measures. Study findings are encouraging, suggesting *In My Shoes* is a feasible and appropriate intervention, and shows promise in improving the self-report school engagement of *all* student participants, as well the classroom participation and subjective school experiences of students on the autism spectrum. Useful insights into ways the intervention and the design of future research can be improved are discussed.

## Introduction

School belonging, bonding, engagement, and attachment – researchers have used many terms over the years to describe the concept of school connectedness [1]. A recent systematic

**Data Availability Statement:** All relevant data are within the paper and its supporting information files.

**Funding:** The author/s received no specific funding for this work.

**Competing interests:** The authors have declared that no competing interests exist.

literature review evaluating the psychometric properties of school connectedness measures thematically categorised factors contributing towards students sense of school connectedness under affective (e.g., feelings of acceptance, inclusion and belonging; feelings of respect and being respected), cognitive (e.g., perceptions of the quality of teacher and peer relationships and support) and behavioural (e.g., actual involvement, participation or engagement; level of effort or persistence or degree of interest or motivation towards school) domains [1]. Collectively, these concepts are critical dimensions of students experience in school and are essential in promoting student development and overall academic success [1]. According to Klem and Connell [2], by high school, 40 to 60 percent of students are persistently disconnected from school in the United States. Research indicates that a sense of school connectedness is an important protective factor to mental and emotional wellbeing [3] and is linked to positive affect, high self-esteem, and life satisfaction [4, 5]. School connectedness has also been found to reduce risk taking and antisocial behaviour and reduce the likelihood of developing depressive symptomatology [6, 7].

Many studies have sought to understand the school experiences of vulnerable or at-risk populations to develop support for these students [7–9]. In recent years, there has been a growth in cross-sectional research exploring the school experiences of students on the autism spectrum. This research indicates that students on the autism spectrum experience significant participation restrictions due to barriers such as lack of teacher and peer understanding of autism and lack of appropriate accommodations, such as modification to the curriculum, and social and physical environments [9, 10]. According to a recent study involving focus groups with educators and parents, participation restrictions can include the following difficulties: remaining calm and in a state for learning in the classroom; building and maintaining relationships; adapting and responding to change and transition throughout the school day; managing conflict in play; and working in groups and engaging in classroom activities and routines [11]. Falkmer and colleagues [12] reported elementary school students on the autism spectrum perceive their participation in mainstream school to be lower than peers and that they are ". . . more bullied, less liked, less involved in interaction, less understood by teachers and more insecure in the school environment compared to peers" [12]. Persistent challenges participating at school can lead to students feeling like they do not belong and are not included in the school environment, which can have significant long-term implications on students' academic, social and emotional wellbeing [7].

Despite evidence emphasising the significant impact school connectedness has on student outcomes, there is an imbalance in the curriculum and a paucity of interventions aimed specifically at increasing students' experience of connection at schools [13, 14], particularly for elementary school students on the autism spectrum. Interventions exist that aim to support students to develop a particular set of skills [15, 16, social skills; 17], with an expectation that these skills will have a flow-on effect on students' participation and inclusion at school [18]. For example, in Australia, the Secret Agent Society (SAS) is a computer game pack and small group program for students on the autism spectrum aged between 8 and 12 years that was developed for use predominately in a clinic setting [19], but has been adapted for use in schools [20]. SAS focuses on improving students' social and emotional skills to help students develop and maintain friendships. However, SAS does not address a range of barriers students on the autism spectrum experience in their participation at school that are specific to the activities, tasks and routines present in the school environment. For example, how to recognise when a peer may be experiencing difficulty in the classroom and strategies that peers can use to help them participate or feel included, or how to manage emotions when things change at school such as when there is an excursion, a sports carnival or a relief teacher. Evidence-based interventions are needed that immerse *all* students in learning that aims to improve students'

interpersonal empathy and ability to display behaviours that help others participate and feel included at school.

This study aimed to evaluate the feasibility, fidelity, and preliminary effectiveness of a curriculum embedded, peer supported, teacher led school-based intervention, entitled *In My Shoes*, with elementary school students on the autism spectrum and their typically developing peers in Western Australia. The development of *In My Shoes*, from conceptualisation to implementation in the school environment, is described in a recently published paper by Hodges et al. [21]. To investigate feasibility (i.e., the impact an intervention has on its end user and the resources required to successfully implement the intervention [22]), we evaluated (a) recruitment capability and sample characteristics; (b) data collection procedures and outcome measures (c) appropriateness (i.e., the extent to which *In My Shoes* is deemed acceptable, satisfying, or appealing to participants); (d) implementation and practicality (i.e., the extent to which *In My Shoes* can be successfully delivered using existing means and resources [23, 24]). To evaluate fidelity (i.e., the degree to which an intervention has been delivered as intended; [25]), we evaluated (e) teacher's delivery of the intervention against specific criteria; (f) parents' receipt and response to weekly parent information handouts; and (g) schools' implementation of whole school activity ideas as recommended in the manual. To explore preliminary effectiveness, we evaluated (h) changes in the classroom participation and subjective experiences of students on the autism spectrum; and (i) students' self-report school engagement and belonging pre-post intervention using a range of outcome measures.

## Methods

### Participants

Grade 3 and 4 independent mainstream classrooms (students aged 8 to 10 years) in the Perth Metropolitan area with at least one student with a confirmed diagnosis of autism or Asperger's syndrome in accordance with DSM-IV [26] or DSM 5 criteria [27], without intellectual disability or severe language impairment, were eligible to participate in the study. Students on the autism spectrum were required to have at least a grade 1 reading level as determined by the Woodcock Reading Master Test – Third Edition to participate [WRMT-III; 28].

### Intervention

*In My Shoes*, is a manualised, peer supported, teacher-led school-based intervention designed to improve the school participation and feelings of connectedness of students on the autism spectrum aged between 8 and 10 years. A number of research activities informed the development of In My Shoes, including: a systematic literature review of the psychometric properties of school connectedness measures [1]; focus groups with parents and educators to explore their perspectives on the school participation of students on the spectrum [11]; a national 2-round Delphi study to gain consensus on the content, delivery and feasibility of the intervention and the application of a theoretical framework to students on the autism spectrum [29]; and regular consultations with a consumer and stakeholder reference group.

*In My Shoes* is designed to be delivered over the course of a school term (approximately 10 weeks) and includes the following components: (1) standardised online professional learning and ongoing face to face or online support for teachers and school leadership staff; (2) teacher-led whole class lesson plans linked to Australian health curriculum; (3) peer training for selected peers; (4) activity ideas to incorporate key messages across the whole school; and (5) weekly parent information handouts and invitations for parents to participate in the intervention. The intended outcomes of *In My Shoes* for *all* students are to:

a. increase understanding and awareness of differences in the way students experience autism and school *(i.e., preferences)*

b. increase feelings of being accepted, respected, included and supported by others in the school social environment *(i.e., school connectedness);*

c. increase self-awareness of strengths and differences and the strengths and differences of peers *(i.e., sense of self);*

d. improve confidence in their abilities to recognise when someone needs help, how to help others and ask for help at school *(i.e., sense of self and activity competence)*; and

e. improve students' interpersonal empathy and use of pro-social behaviours to include peers in the classroom and playground *(i.e., activity competence).*

Each whole class lesson plan is designed to target *specific* intervention outcomes. Some lesson plans focus on targeting one intervention outcome, whereas others target several intervention outcomes. Over the 10 lesson plans, all intervention outcomes are targeted several times using a range of evidence-based intervention techniques including role play and video modelling, as well as educational practices identified to be feasible by educators (e.g., worksheets, whole class discussion [30]. The core concept of the whole class program, '*look*, *think*, *decide*', teaches perspective taking and social problem-solving skills by helping students to recognise body clues and how to use these to deduce what someone else might be thinking and feeling so that they can decide on the best course of action to help peers participate and feel included. Students are asked regularly throughout the program to reflect, using interactive video resources and comic-strip style illustrations, on what they would think or how they would feel if they were in a particular character's shoes and what they think the character should do to support their peers in different situations. Each lesson aims to teach these skills with a particular context in mind; for example, how to recognise and support peers in the classroom versus the playground versus school organised events such as excursions, assemblies, or sports carnivals. Intervention resources are made available to schools on a USB memory stick and include professional learning video presentations, an online interactive PDF manual, printable lesson plans, worksheets and resources, and interactive video resources with real-life students on the autism spectrum sharing their school experiences. Refer to Hodges et al. [21] for more information about the conceptualisation of the intervention and the content and delivery of each intervention component, including a schematic overview of whole class lesson topics.

## Procedures and measures

Prior to conducting the study, ethics approval was obtained from the Human Research Ethics Committee at Curtin University (HREC 2016-0150) and research was approved by Association of Independent Schools Western Australia (AISWA) and Catholic Education Western Australia.

The primary researcher sent a participant information sheet via email to the principals of all AISWA and Catholic mainstream elementary schools in the Perth Metropolitan area. The primary researcher followed up with a phone call to identify schools that were willing and eligible to participate. Once written informed consent and assent was obtained from students on the autism spectrum, their parents and teachers via the school principal, the primary researcher contacted teachers and parents to answer any questions, organise screening assessments and the collection of pre-intervention data. The school principal and classroom teacher were responsible for informing parents of typically developing students regarding the schools involvement in the research. Written informed consent was only obtained from parents of

typically developing students that completed the Home & Community Social Behaviour Scale [HCSBS; 31]. Teachers were given access to intervention resources on a USB and instructed to complete an online professional learning package located on the USB prior to the intervention. The resources included four short video presentations ranging between 4 and 24 minutes of the primary researcher explaining the intervention and providing practical demonstrations of intervention techniques such as video modelling. School leadership staff involved in supporting teachers delivering the program (e.g., deputy school principals, school psychologists or learning support coordinators) were encouraged to complete the professional learning so that they were able to adequately support teachers and help to implement the whole school component of the intervention. The primary researcher then arranged follow up online or face-to-face meetings with teachers and school leadership staff to clarify any components of the intervention and to help teachers specifically apply concepts to their classroom. Teachers then delivered the whole class program across Term 3 (July to September 2020), usually delivering one 45-minute lesson per week over 10 weeks.

## Screening assessments

Screening assessments were conducted pre-intervention to identify and describe the skills and abilities of students on the autism spectrum and to confirm their eligibility for the study. Participants were informed that if they did not meet eligibility criteria, they could still participate in the study, however, their data would not be used. All participants that expressed interest in participating, met eligibility criteria and data were included in the study. Socio-demographic information was collected from schools, teachers, and parents of students on the autism spectrum. Socio-economic status was determined using the Socio-Economic Indexes for Areas (SEIFA) [32] and students' diagnoses were confirmed via school and parent report.

The teacher report Children's Communication Checklist Second Edition [CCC-2; 33] was used to screen students' expressive and receptive language skills. Items (e.g., stands too close to other people when talking to them) are rated on a four-point scale (e.g., 0 = never, 3 = several times per day) to indicate frequency of occurrence of various communication behaviours. The CCC–2 has a high level of sensitivity and specificity in identifying students on the autism spectrum or pragmatic language impairments [33]. The Woodcock Reading Master Test-Third Edition [28] was used to screen reading comprehension to confirm that students had at least a Grade 1 reading level to be able to respond to survey questions.

Teachers' self-efficacy can impact on their ability to deliver school-based interventions and provide support to students on the autism spectrum. The 30-item Bandura's Teachers Efficacy scale was therefore used to assess teachers' efficacy beliefs to identify if this could be a confounding variable impacting study findings [34]. Items (e.g., How much can you do to motivate students who show low interest in schoolwork?) are anchored on a nine-point scale ranging from 'nothing, very little, some influence, quite a bit, and a great deal'. A higher score indicates greater efficacy. In the current study, mean teacher efficacy was computed. The average score for the 30-item score had strong internal consistency, with Cronbach's α values of 0 0.95 in the current study.

## Proximal outcome measures

**Feasibility.**   Information related to the feasibility of the intervention were gathered using a combination of quantitative and qualitative methods. All students participating in the study completed a paper-based feedback survey in the final week of the intervention. Consent from all parents of students in participating classrooms was obtained at the schools' discretion. The survey asked students to respond to statements about the intervention such as "I enjoyed *In*

*My Shoes*" and "*In My Shoes* activities were interesting" using a 4-point Likert scale (1= strongly disagree to 4 = strongly agree). Teachers, parents, and school leadership staff involved in supporting teachers were sent a link to an anonymous online feedback survey, individual-ised to their role, post intervention. These surveys asked participants to respond to statements such as "*In My Shoes* was a positive experience" and "the content of In My Shoes was relevant" using a 5-point Likert scale (1 = strongly agree to 5 = strongly disagree). Participants were prompted to provide reasoning if they responded 'neutral', 'disagree' or 'strongly disagree' to any of the statements. Responses to feedback surveys were supplemented with qualitative interview data through specific lines of questioning relating to the implementation and practi-cality of the intervention.

**Fidelity.**   The fidelity protocol for this study was based on the behaviour change consor-tium treatment fidelity recommendations [25] (see S1 Table). The primary researcher observed one lesson in each classroom and scored teachers on a fidelity checklist, which included questions relating to adherence, duration, quality of delivery, student responsiveness and programme specificity (i.e., whether teachers adhere to activities as designed and show knowledge of content and intervention strategies). Teachers were also required to complete weekly online fidelity checklists to ensure intervention was being delivered as stated in manual. Parents' receipt and response to weekly parent information handouts was evaluated via an online feedback survey post-intervention and following interviews. Schools' adherence to the whole school component of the intervention (e.g., implementation of whole school activity ideas such assembly items, newsletter inserts) was evaluated via online teacher and school lead-ership feedback surveys and in interviews with teachers.

**Preliminary effectiveness.**   To evaluate the effectiveness of the intervention in targeting identified intervention outcomes (see page 7), several measures were administered pre-post intervention. Some of these measures were conducted with students on the autism spectrum only and others with all students participating in the study.

**Classroom participation.**   The Behaviour Assessment System for Children – Third Edi-tion Student Observation System [BASC-3 SOS; 35] was used to conduct direct observations of the classroom behaviour of students on the autism spectrum. The SOS uses the technique of momentary time sampling (i.e., systematic coding during three second intervals spaced 30 sec-onds apart over a 15-minute period) to record a range of student behaviours including positive (e.g., teacher-student interaction) and negative behaviours (e.g., inappropriate movement or inattention). It also includes a 71-item observer rating scale that is completed after the time sampling procedure that gives in-depth information about students' behaviours that may impede or promote learning and adjustment in the classroom (Part A). Two 15-minute video recordings of students on the autism spectrum, participating in similar classroom activities, were taken in the first and final week of the intervention and sent to the primary researcher. The primary researcher recorded students' behaviour on the BASC-SOS. An independent rater who is a qualified occupational therapist with experience working in schools and with students on the autism spectrum, was trained in the use of the SOS and scored a 40% random sample of the pre-post video observations to establish inter-rater reliability. The independent rater was blinded to all aspects and purposes of the study, and scored students on the BASC-3 independently to the primary researcher to minimise bias.

**Subjective school experiences.**   There is not one single measure that adequately captures all factors that contribute to students sense of school connectedness. Therefore, a battery of measures was used to evaluate self-reported changes in this construct and related intrinsic fac-tors (i.e., preferences, sense of self, activity competence) pre-post intervention. Due to various terms used to describe the concept of school connectedness (e.g., belonging, bonding,

engagement), we have collectively labelled these measures under the heading 'subjective school experiences' for the purposes of this paper.

Experience Sampling Method (ESM) was used to evaluate changes in the school connectedness of students on the autism spectrum, by exploring the nature and quality of their experience while participating at school. ESM, an 'in-the-moment' technique that is ". . .commonly used for the examination of the context and content of individuals' daily life from their own perspective" [36] and has been found to be a reliable and valid tool to self-report the participation experiences of children on the autism spectrum aged 8 to 10 years [36]. This methodology was chosen as it captures the influence of context on experiences, which allows for the examination of individual values relating to school participation and identifies fluctuations in perceptions of everyday experiences [36]. Collecting ESM data at multiple moments throughout the day also minimises error due to recall, distortion and rationalisation and allows for exploration of the dynamic relationship between subjective experiences and everyday contexts [37, 38].

Students on the autism spectrum were required to complete a survey that had been loaded onto the mEMA app [39], which is an ESM platform designed for IOS devices. The mEMA app prompts participants to complete the ESM survey, time stamps the response and stores data for analysis. The survey was adapted from a version developed by Chen and colleagues [40] and included closed questions and scaled items related to the student's participation in school occupations, including the specific place (e.g., where were you when you were beeped?), the specific activity (e.g., what was the main thing you were doing?) and interaction status (e.g., who were you with? Were you talking with someone? Who were you talking to?). The ESM survey also explored the quality of their experiences relating to enjoyment, difficulty, interest, degree of involvement and importance. Emotions were explored on a continuous scale across five domains: anxious-relaxed, lonely-sociable, sad-happy, angry-friendly, and bored-excited. The format of the questions included multiple choice, yes/no and visual analogue scales for items relating to emotions. The mEMA platform was chosen over others as it allows researchers to use images to supplement text as a visual support for students on the autism spectrum.

The ESM survey was piloted with two typically developing students aged between 7 and 11 years to ensure the questions were clear and developmentally appropriate, and the device was easy to use. Adjustments were made to the survey based on observations and students' feedback. The Flesch Kincaid readability test [41] showed that the grade level of survey questions was 3; lower than participants reading comprehension levels identified by the WRMT-III. Prior to data collection, students on the autism spectrum, their parents and educators were provided training in the use of the device and the mEMA app. Training involved the researcher asking students to read and respond to survey items, clarifying questions and troubleshooting students' responses.

Students were provided with an iOS phone with the mEMA app installed in the first and final week of the intervention. The device randomly prompted students to respond to the survey 5 times a day for a week between 7am and 5pm. This sampling period enabled researchers to capture students' experiences while at school, as well as their transition to-and-from school. Students were informed that they could skip prompts that occurred at inconvenient times; that they would be reminded three times every five minutes to complete the survey and that the survey would become inactive after 15 minutes. Students were encouraged to seek help from their parents, educators or contact the researcher directly if they required assistance during sampling periods.

All students in participating classrooms were required to complete a battery of outcome measures, including the Student Engagement Instrument – Elementary Version [SEI-E; 42], Belonging Scale [43] and four scales developed by the research team.

The SEI-E and Belonging Scale were used to evaluate changes in students school connectedness. The SEI–E assessed students' self-report levels of cognitive and affective engagement in school. The SEI–E includes 31 items and four subscales (i.e., teacher student relationships, peer support for learning, future goals and aspirations and family support for learning); scored on a 5-point Likert scale (1 = strongly disagree, 5 = strongly agree), with higher scores indicating a higher level of engagement. For example, 'other students here like me the way I am', 'students at my school are there for me when I need them' and 'teachers at my school care about the students'. The SEI–E was adapted from Appleton and colleagues [44] original SEI to ensure items addressed all relevant engagement constructs and were developmentally appropriate for primary school students. The SEI was found to have the strongest psychometric properties of 15 measures in a recent systematic review [1]; with the SEI–E showing promising psychometrics from preliminary studies for students in years 3 to 5 [42, 43].

The Belonging Scale assessed students' sense of school belonging. It is a 12-item adapted version of the Psychological Sense of School Membership scale [45] designed for use with students from 8 years of age to. The 12 items include six that focus on students' general feelings towards school and sense of belonging (e.g., I feel really happy at my school) and six that focus on their perception of support, help and acceptance from adults and peers at school (e.g., there is an adult in school I can talk to about my problems). Students respond using a 3-point Likert scale (1= no not true, 2 = not sure, 3 = yes). The Belonging Scale has been validated with students aged 8 to 11 years and has a Cronbach's alpha of 0.79, within the range of values ($\alpha$ = 0.77 – 0.88) commonly reported for the 18-item Psychological Sense of School Membership [45].

Four-scales developed by the research team, entitled 'In My Shoes', 'in the past week', 'involvement' and 'learning about the autism spectrum', were used to evaluate changes in students' interpersonal empathy, self-perceived confidence and involvement at school, and understanding of autism (see intervention outcomes listed on page 7).

In My Shoes, was a situation-based scale that presented 10 social situations that commonly occur at school. Students were required to select how they would respond to each social situation from a multiple-choice list (e.g., you are playing a game of four square with your friends. You see Johnny is sitting on his own in the playground. Do you: A: ask Johnny to come and play; B: ignore Johnny. He's not good at four square or C: leave Johnny alone. You know that he likes playing by himself). The purpose of the scale was to assess changes in a student's ability to identify pro-social behaviours that would lead to the inclusion of their peers (see intervention outcome (e) on page 7). Multiple choice responses that involved higher level of interpersonal empathy were scored higher (e.g., 2 = ask Johnny to come and play; 0 = ignore Johnny. 1 = leave Johnny alone). Each social situation was directly related to intervention lesson content.

'In the past week' included 12-items that assessed students' self-perceived confidence in: asking for help, knowing when a peer needs help, helping a peer, encouraging a peer, inviting a peer to play, starting, or joining in conversation and sharing with a peer; all skills that were targeted in the intervention (see intervention outcome (d) and (e) on page 7). Items were rated on a 4-point Likert scale (1 = not confident at all to 4 = very confident), with higher scores indicating higher levels of confidence.

The 'involvement' scale included 8-items assessing students' self-perceived involvement in classroom, school, and extracurricular activities. Items (e.g., most mornings, I look forward to going to school, I work hard at school, and I am an active participant in classroom activities)

were scored on a 3-point Likert scale (1= no not true, 2= not sure, 3=yes true), with higher scores indicating higher levels of self-perceived involvement.

Finally, 'learning about the autism spectrum' included eight statements about autism that students were required to identify as true or false (e.g., children on the autism spectrum brains work differently and only boys have autism). This scale was administered at the beginning and end of the second lesson, to evaluate changes in students understanding of autism (see intervention outcome (a) on page 7). This lesson focused on increasing students understanding of autism using a documentary style video of students on the autism spectrum. All self-developed questionnaires were reviewed by a speech therapist for language comprehension and trialled with two typically developing elementary school students.

Semi-structured interviews were conducted by the primary researcher with teachers, parents, and students on the autism spectrum pre-post intervention to verify and enrich quantitative data. Interview guides were specifically designed to gather more information about the feasibility and perceived benefits of the intervention from different participants perspectives. For example, teachers were asked questions like 'how easy was it to implement *In My Shoes* in your classroom?', and 'do you think peers have experienced any benefits as a result of participating in *In My Shoes*? If so, can you please share some specific examples with me?'

### Distal outcome measures

**Preliminary effectiveness.** The parent-report Home & Community Social Behaviour Scale (HCSBS; 31) and the teacher-report School Social Behaviour Scale [SSBS; 46] were used to describe and evaluate changes in the social competence and behaviour (see intervention outcome (e) on page 7) of students on the autism spectrum in the home and school environment. The HCSBS has excellent internal consistency (social competence, $\alpha = 0.96$; antisocial behaviour, $\alpha = 0.98$) and good to excellent ($\alpha = 0.82 - 0.91$) test-retest reliability [31]. The SSBS has excellent internal consistency (social competence, $\alpha = 0.91$; antisocial behaviour $\alpha = 0.98$) and acceptable ($\alpha = 0.68 - 0.80$) test-retest reliability [46]. A maximum of five parents of typically developing students from each classroom were asked to complete the HCSBS for their child, to evaluate differences and changes in samples pre-post intervention.

### Data analysis

Data were analysed using the Statistical Package for the Social Sciences (SPSS Version 27) software. Descriptive statistics were used to summarise the profiles of participants. Non-parametric Wilcoxon signed-rank and Mann-Whitney U independent samples tests were used to compare data pre-post intervention. To determine inter-rater reliability of the BASC-SOS; MedCal (Version 19.6.1) was used to conduct a weighted kappa for Part A and SPSS was used to calculate an intra-class correlation coefficient for Part B. Hierarchical linear modelling (HLM) was attempted with ESM data to explore casual links between the intervention and students subjective school experiences. Given that semi-structured interviews were designed to gather specific information about the intervention's feasibility and effectiveness; interviews were analysed using content analysis and data were grouped into subheadings relating to key areas of focus for feasibility studies as outlined by Bowen and colleagues [24]. Credibility was improved through researcher triangulation, peer debriefing and member checking to test findings and interpretations with participants [47]. Transferability was met through the provision of detailed descriptions of participants and of results [48, 49]. Dependability was enhanced through use of an audit trail, field notes and reflexive journal [50] and confirmability through a description of the methodology used to analyse, organise, describe and report on themes within the data [47, 48, 51–53].

## Results

### Feasibility

**Recruitment capability and sample characteristics.** *Descriptive characteristics of students on the autism spectrum.* Ten students on the autism spectrum aged between 8 and 10 years participated the study. Most students had at least one sibling and 90% of students were male and the only child in their family with a diagnosed disability. Ninety percent of students had an additional diagnosis with anxiety and ADHD being reported in more than 50% of the sample. Half of students changed schools at least once due to parent reports of inadequate support or bullying at their previous school. All student participants were on an Individual Education Plan and had access to an Education Assistant. All students accessed services outside of school including occupational therapy, speech therapy and psychology. All students had at least a grade 1 reading level so were able to comprehend survey items (see Table 1).

*Descriptive characteristics of participating teachers and schools.* Eight teachers represented six mainstream independent co-educational elementary schools in the Perth metropolitan area. Four of the teachers had more than 10 years teaching experience, with only one teacher newly graduated. All teachers had experience teaching students on the autism spectrum and most teachers had experience working in other grade levels (70%) and schooling sectors (50%). The average SEIFA decile was 9.3 (SD: 1.06; range: 7 – 10). A high SEIFA decile reflects a relative lack of disadvantage rather than relative advantage; for example, few households with low incomes, few people with no qualifications or in low skilled occupations [32]. As shown in Table 2, most schools came from higher decile regions of Western Australia and were large in student size. Only one teacher taught in a classroom with less than 25 students. Five out of eight teachers reported moderate to high levels of teaching self-efficacy.

Examining recruitment capability and resulting sample characteristics was important in determining whether *In My Shoes* was relevant to study participants and if future efficacy studies would be successful [23]. Students on the autism spectrum appeared to have characteristics

**Table 1. Characteristics of students on the autism spectrum.**

| Student characteristics (n=10) | Mean | SD | Range | Percentile |
|---|---|---|---|---|
| **Child age (years)** | 8.8 | 0.63 | 8-10 | - |
| **WRMT-III** | | | | |
| Word comprehension (grade) | 3.59 | 1.01 | 1.9-5.5 | - |
| Passage comprehension (grade) | 2.50 | 0.87 | 1.6-4.3 | - |
| **CCC-2** | | | | |
| Speech | 6.90 | 4.04 | 0-12 | 22 |
| Syntax | 6.30 | 3.47 | 1-12 | 15 |
| Semantics | 5.30 | 2.91 | 1-11 | 6 |
| Coherence | 4.60 | 3.17 | 2-13 | 8 |
| Inappropriate initiation | 5.50 | 2.01 | 3-9 | 10 |
| Stereotyped language | 4.10 | 1.60 | 2-7 | 3 |
| Use of context | 3.10 | 1.79 | 0-7 | 1 |
| Nonverbal communication | 2.90 | 1.73 | 1-7 | 1 |
| Social relations | 2.90 | 2.77 | 0-9 | 2 |
| Interests | 4.10 | 1.20 | 2-6 | 1 |
| General communication composite | 38.70 | 15.38 | 22-75 | 3 |
| Social interaction deviance composite | -7.70 | 10.81 | -28-6 | |

*Note.* CCC-2 Children's Communication Checklist 2<sup>nd</sup> edition; WRMT-III, Woodcock Reading Master Test (required Grade 1 reading level).

**Table 2. Characteristics of teachers and schools.**

| Teacher (n=8) and school (n=6) characteristics | Frequency (%) |
|---|---|
| **Mean school SES** | |
| 1 – 6 (lower decile range) | 0 (0) |
| 7 – 8 (mid decile range) | 2 (33.3) |
| 9 – 10 (upper decile range) | 4 (66.7) |
| **School size based on total number of students** | |
| Small (<375 students) | 1 (16.7) |
| Mid-range (375-975 students) | 1 (16.7) |
| Large (>975 students) | 4 (66.7) |
| **Classroom (n=8) size** | |
| Small (<25 students) | 1 (12.5) |
| Mid-range (25 – 30 students) | 5 (62.5) |
| Large (>31 students) | 2 (25.0) |
| **Self-efficacy in teaching** | |
| Low quartile | 2 (28.6) |
| Middle half | 3 (42.9) |
| High quartile | 2 (28.6) |

that were consistent with what was reported literature as students who would be appropriate for the intervention. While the recruitment process was time consuming, we were able to recruit an adequate sample for the purposes of the feasibility study. Several schools expressed interest in participating, however, parents of students on the autism spectrum declined to participate mostly because their child was not aware of their diagnosis. Schools that declined to participate in the study attributed this to not meeting eligibility criteria, lack of time and resources, and pressure in meeting curriculum requirements due to COVID-19 school closures in the previous term.

**Data collection procedures and outcome measures.** Teachers reported several challenges relating to the collection of data during the feasibility study. Data were collected in the first and final week of term, which are often the busiest, most unstructured weeks of term which can be highly charged with emotion for students. Teachers felt students may have experienced fatigue, altered their responses due to the teacher's presence, and were concerned students' responses may have varied depending on the time of day. Maria (teacher) said,

> It would depend on the kind of day they are having, and this can change so quickly. . .even the time of day you do it, they might have had a fight with mum before they got out of the car. You never know with kids. You could literally do the same thing on two different days and they get totally different answers.

The timing of data collection during this study may have impacted results and emphasises the importance of strategic timing of data collection when conducting research in schools. Authors suggest future pilot studies consider collecting data mid-term to mid-term to minimize the impact of these contextual factors on study findings.

Overall, ESM proved a useful tool in capturing students' in-the-moment lived experiences at school. Students on the autism spectrum reported that they found it easy to use the ESM device and complete the surveys. Most students reported enjoying the responsibility of having the device and responding to ESM surveys. Only one student reported that she did not like the attention the device brought to her in the classroom. The primary researcher supported this

student and school to implement strategies to minimize student anxiety. Students' response rate to ESM surveys was relatively stable over time, reducing slightly from an average of 75.5% at pre-intervention to 72.8% post-intervention. Some participants reported that their device did not prompt consistently every day, which may have impacted response rate. The primary researcher supported these students and schools to troubleshoot technical issues and substituted these survey instances with paper-based surveys to maximize response rates.

**Appropriateness.** Overall, the intervention was well received with most teachers, school leadership staff, and parents reporting *In My Shoes* was a positive experience and that it was relevant, important, and beneficial to students on the autism spectrum and their peers (see S2 Table). Given the intervention only targeted students in grade 3 and 4 and ran over a term, it is a positive outcome for the feasibility of the intervention, that 25% of school leadership reported the intervention made sustainable changes to the whole school and half of teachers thought it increased the participation of students on the autism spectrum. Most teachers (87.5%), school leadership (100%) and parents (90%) reported they would recommend the intervention to another teacher or school. Students' feedback was overwhelmingly positive with more than 80% of students reporting they enjoyed participating in *In My Shoes* and that it was fun, interesting, made sense to them, and that they learnt something new (see S3 Table). Qualitative comments from 200 student feedback surveys highlighted a preference for 'hands on' activities such as role play; suggested minimizing time spent sitting on the mat engaging in whole class discussion, and completing surveys related to data collection.

Qualitative data revealed that the intervention has several strengths. Teachers reported experiencing several benefits including improved understanding of autism, ways to support students in the classroom, and their ability to reflect on their teaching and practice. Amanda said,

> To be honest there is only so much you can learn at university, your lessons were able to sort of break it all down for me – like how to support someone who has autism or someone who struggles with change – just to know some more strategies for myself has been very helpful.

Teachers valued the 'ready-to-go' nature of intervention resources including a detailed interactive PDF manual, online professional learning, and video resources. Teachers reported students engaged particularly well with lessons that involved role play and video modelling, benefited from access to the power-point resources as an additional visual support, and related well to the diversity of characters presented in lesson plans. These findings were supported in student interviews; students added by suggesting researchers incorporate more technology into lesson plans using game-based learning platforms (e.g., Kahoot) and iPads to complete worksheets to maximize student engagement.

Overwhelmingly, a significant benefit of the program has been increased peer understanding and acceptance of autism, according to parents and teachers. Teachers reported specific examples of instances where post intervention students had recognised when a peer (with or without autism) needed help and actively supported peers in the classroom or playground. For example, Lachlan (teacher) said,

> She's had meltdowns in the past and the kids sort of just stared. They did not know what to do. They were worried. Whereas now if she starts blocking her ears, there's been a few occasions where they've come to me and said I think Jessica needs a break. That is straight from the program. She benefits because the kids know how to better help her. . .the kids have benefit because they have a better knowledge and understanding and so do I.

Jess (teacher) described an incident at lunch where a student on the autism spectrum was standing on their own and the peers approached the student and asked him if he wanted to join in, saying "I don't know if that would have happened without the program to be honest".

**Implementation and practicality.**   Teachers reported some 45-minute lesson plans took longer than expected to deliver (mean: 65 minutes; range: 30 – 90 minutes). Jessica said, "The biggest issue is just, it's time. By the time we do protective behaviours and everything else, it is finding the time. Some of the lessons were spot on, some of them were too long." Teachers acknowledged, however, time management is highly dependent on teachers' skills and experience. Teachers suggested condensing content to enable teachers to deliver the lesson within a 45-minute time frame, simplifying worksheets and, in some cases, substituting worksheets movement-based activities to maximize student engagement.

Teachers reported challenges implementing the whole school and parent component of the intervention, attributing this to a lack of time and resources, lack of priority placed on health in the curriculum, and COVID-19 restrictions. At the time of the pilot, due to COVID-19 restrictions, parents were not permitted to be onsite at schools and whole school events were limited due to social distancing requirements. School events that had been cancelled in the previous term had been pushed forward, which limited time available for whole school activities. Loretta said, "I think it was in the too hard basket, to be honest. COVID has had a lot to do with it and it just kind of got put on the back burner. . . it's not the top priority". Most teachers recommended that to make an impact at a whole school level, the intervention needs to target more grade levels so that there is common terminology and a shared understanding within the school.

Teachers acknowledged the importance of parent involvement but were doubtful about parent uptake even if COVID-19 was not a barrier. Toby (teacher) said,

> To be fair I email parents lots of things that they do not even read. I feel like even if it was less, there is just a chunk of parents that are going to read it and there is a chunk that won't, but I think it has the potential to really get the parents involved in a very positive way.

Maria (teacher) said,

> At this particular school, I honestly think health would be brushed off as something that is not the most important subject and certainly not one where parents would feel like they need to come on their workday. . . especially if they think my kid's not autistic so it doesn't apply to me.

Parents of students on the autism spectrum expressed frustration that information was not sent home as they felt it limited their ability to generalize their child's learning and identify whether changes in their child's behaviour were due to the intervention or other reasons. Parents expressed a desire to be informed and involved in their child's learning, but acknowledged the way in which this is implemented needs to be realistic. Jackson (parent) said,

> There are some parents who are barely surviving themselves. And it is just a case of pushing the kids out the door. . . you can throw as much information at them but that is not going to get in because they can't even take care of themselves. And there's other people which are probably that proactive, that anything that you bring up, they have probably already considered because they like to be ahead of the curve.

Teachers suggested adapting the program to include children with other neurodiversity's, such as ADHD, to make the intervention more applicable to a broader student population, which may increase parent buy in and uptake of the intervention at a whole school level. Parents and teachers also suggested condensing written parent information and utilizing other forms of media (e.g., uploading work pieces or videos to school portal) where possible to maximize parent engagement.

**Fidelity.** Teachers reported a 25% improvement in confidence in implementing *In My Shoes* after they completed standardized online professional learning; with 87.5% of teachers reporting, they were 'fairly confident' or 'very confident' in delivering the intervention in their classroom. There was a 98.8% response rate to online fidelity surveys, which took teachers approximately 4 minutes to complete every week. Teachers reported sending parent information handouts home 74% of the time; attributing this to lack of time and school policies relating to the amount of information that can be sent to parents. Teachers reported conducting activities in the lesson plan as specified in the manual 90% of the time and that students were actively engaged in lessons more than 95% of the time. The primary researcher observed at least one lesson in every classroom, either in person or via video recording. Teachers were observed to deliver lesson plans as specified in the manual on average 90.9% of the time (range: 77.8 – 100%).

Only fifty percent of parents reported reading parent information handouts, with the remaining parents reporting that they did not receive information from their school. The majority of school leadership staff reported their school did not implement whole school activity ideas due to lack of time and resources and COVID-19 restrictions.

**Preliminary effectiveness.** *Changes in the classroom participation of students on the autism spectrum reported on the BASC-3 SOS.* There was a positive trend in student behaviour in the classroom including increased peer interactions and responsivity to their teacher and less inattentive behaviours (see Tables 3 and 4). Students were observed to display significantly less inattentive behaviours and were observed to talk more to peers post intervention. Qualitatively, there were more reports of peers prompting students on the autism spectrum (e.g., to re-engage in an activity, to locate materials, or to help complete a task) and of students appearing happier and more engaged during observations post intervention. Inter-rater reliability was deemed excellent with a weighted kappa of 1.0 (100% agreement) for Part A and an intra-class correlation coefficient greater than 0.90 for all sections in Part B (i.e., $\alpha > 0.90$ response to teacher, $\alpha = 0.992$; peer interaction, $\alpha = 0.997$; work on school subjects, $\alpha = 0.996$; transition, $\alpha = 0.998$; inappropriate interactions, $\alpha = 0.963$; inattention, $\alpha = 0.994$; inappropriate vocalisations $\alpha = 0.999$).

*Changes in the subjective school experiences of students on the autism spectrum reported using ESM.* There was a statistically significant reduction in students on the autism spectrum reporting difficulties in the classroom post intervention. Several ESM findings trended in a positive direction but did not reach significance. For example, students reported higher levels of enjoyment; needing less help; being with classmates more; finding classwork less difficult; being more interested in classwork; and feeling more sociable and excited when in the classroom post intervention. When students reported that they needed help, classmates helped them more post intervention. Students also reported feeling increased enjoyment and interest and feeling more sociable when with their teacher and reported feeling happier when listening to their teacher post intervention, but the change was not statistically significant.

Some results relating to students' emotions were inconsistent. Although students reported feeling more interested in classwork, they also reported feeling more worried when participating in classwork post intervention. Students also reported increased enjoyment when with

**Table 3. Difference in BASC-3 SOS Part A observations pre-post intervention for students on the autism spectrum.**

| Item | Pre (Median, IQR) | Post (Median, IQR) | P value |
|---|---|---|---|
| **Response to teacher** | | | |
| Listening to teacher/ classmate or following directions | 3 (0) | 3 (0) | 0.771 |
| Interacting with teacher in class/ group | 2 (1) | 2 (2) | 0.336 |
| Working with teacher one on one | 1 (1) | 1 (1) | 0.588 |
| Standing at teachers' desk | 1 (0) | 1 (0) | 1.000 |
| **Peer interaction** | | | |
| Playing/working with other students | 1(1) | 2(2) | 0.065 |
| Talking with other students | 1(1) | 2(1) | 0.009* |
| Touching another student appropriately | 1(0) | 1 (1) | 0.194 |
| **Working on school subjects** | | | |
| Doing seat work | 3(2) | 3(2) | 0.857 |
| Working at a computer or workstation | 1(0) | 1(0) | 0.607 |
| Other | 3(1) | 3 (0) | 1.000 |
| **Transition movement** | | | |
| Putting on/taking off coat | 1(0) | 1(0) | 1.000 |
| Moving around room (appropriately) | 2(1) | 1(0) | 0.513 |
| Preparing materials for beginning/end of lesson | 2(1) | 2(0) | 0.204 |
| Being out of the room | 1(0) | 1 (0) | 0.792 |
| **Inappropriate interactions** | | | |
| Preventing others from working | 1(0) | 1(0) | 0.607 |
| Ignoring appropriate requests from others | 1(0) | 1(0) | 0.667 |
| Distracting others by intruding into others personal space | 1(0) | 1(0) | 0.607 |
| Distracting others by touching (nonsexual) | 1(0) | 1(0) | 0.607 |
| Distracting others by making noise | 1(0) | 1(0) | 0.989 |
| Distracting others by moving around | 1 (0) | 1 (0) | - |
| **Inappropriate movement** | | | |
| Fidgeting in seat | 2 (1) | 1 (2) | 0.057 |
| Walking around classroom | 1 (0) | 1 (0) | 0.461 |
| Using electronic device | 1 (0) | 1 (0) | 1.000 |
| Being removed from the classroom | 1 (0) | 1 (0) | 1.000 |
| Using work materials inappropriately | 1 (0) | 1 (1) | 0.627 |
| Passing notes | 1 (1) | 1 (0) | 1.000 |
| Copying answers | 1 (0) | 1 (0) | 0.792 |
| Jumping out of seat | 1 (0) | 1 (0) | 0.728 |
| Running around classroom | 1 (0) | 1 (0) | 1.000 |
| Sitting/standing beside desk | 1 (0) | 1 (0) | 0.967 |
| Sitting/standing on desk | 1 (0) | 1 (0) | 0.792 |
| Clinging to teacher | 1 (0) | 1 (0) | 1.000 |
| **Inattention** | | | |
| Staring blankly/ daydreaming | 3 (1) | 2 (1) | 0.070 |
| Doodling | 1 (0) | 1 (0) | 0.588 |
| Looking around | 3 (1) | 2 (0) | 0.013* |
| Looking at hands | 1 (2) | 1 (0) | 0.095 |

(*Continued*)

**Table 3.** (Continued)

| Item | Pre (Median, IQR) | Post (Median, IQR) | P value |
|---|---|---|---|
| Fiddling with objects/ fingers | 3 (1) | 1 (2) | 0.008* |

*Note.* BASC-3 SOS – Behaviour Assessment System for Children Student Observation System 3[rd] edition

*p<0.05

Part A – ordinal scale

1= not observed

2= sometimes observed

3= frequently observed; inappropriate vocalisations, somatisation, repetitive motor movements, aggression, self-injurious behaviours, inappropriate sexual behaviour and bowel/bladder problems were not observed and therefore not included in Part A data.

their classmates', but at the same time reported feeling angrier when with their classmates' post intervention. Refer to S4 and S5 Tables for differences in ESM data pre-post intervention.

Benefits to students on the autism spectrum were reported in interviews. Parents and teachers reported increased: student self-awareness of their diagnosis and differences; feelings of self-confidence and empowerment; peer connections and sense of belonging. There were also reports of less friendship challenges and improved social (e.g., ability to join in a game and work in groups) and self-regulation skills. One of the teachers, Maria said,

> He seems more confident in himself and the fact that people weren't thinking that because he couldn't do stuff or that he got upset easily was because there was something wrong with him. . . the fact that the whole class had an understanding of [autism] and were openly talking about it and accepting of it, made him feel more confident.

Amanda (parent) said, "Its boosted his confidence. . . it has really made him feel more accepted and that it's okay to be a bit different". These notions were supported in interviews with some students on the autism spectrum reporting an increased sense of confidence and feelings of empowerment when sharing their experience of autism with their peers. Some students also reported that they had formed new friendships and that their peers seemed "a little

**Table 4. Difference in BASC-3 SOS Part B observations pre-post intervention for students on the autism spectrum.**

| Item | Pre Mean (SD) | Post Mean (SD) | Z score | P value |
|---|---|---|---|---|
| Response to teacher | 3.200 (2.275) | 3.650 (2.981) | -0.119 | 0.906 |
| Peer interaction | 2.450 (3.201) | 4.450 (4.693) | -1.807 | 0.071 |
| Work on school subjects | 17.150 (5.53) | 16.200 (8.131) | -0.153 | 0.878 |
| Transition movement | 2.250 (1.961) | 3.650 (2.698) | -1.897 | 0.058 |
| Inappropriate interactions | 0.050 (0.158) | 0.600 (1.266) | -1.089 | 0.276 |
| Inappropriate movement | 1.900 (4.040) | 0.100 (0.316) | -1.841 | 0.066 |
| Inattention | 10.950 (6.985) | 5.250 (5.313) | -2.077 | 0.038* |
| Inappropriate vocalisations | 0.100 (0.316) | 0.050 (0.158) | -0.447 | 0.655 |
| Other | 0.500 (1.414) | 1.300 (2.123) | -1.625 | 0.104 |

*Note.*

BASC-3 SOS – Behaviour Assessment System for Children Student Observation System 3[rd] edition

*p<0.05

Part B – continuous scale reporting observed counts of behaviour; somatisation, repetitive motor movements, aggression, self-injurious behaviours, inappropriate sexual behaviour and bowel/bladder problems were not observed in the sample and therefore not included in Part B data.

nicer" in that they would play with them more in the classroom and playground post intervention. In some instances, teachers and parents reported increased student participation in the classroom and other school related activities such as assemblies and extra-curricular sport.

*Changes in the subjective school experiences of students across the sample.* Students across the sample reported statistically significant higher levels of engagement and intrinsic motivation at school post intervention. Students also reported improved peer support while learning, but this was not found to be statistically significant. SEI-E scores at or below the 10th percentile are most significant indicators of low student engagement. Pre-intervention, students who scored 89 or lower were deemed at risk of low engagement. Post intervention, students who scored 93 or lower were deemed at risk of low engagement. The increase in the cut off for the 10th percentile from 89 to 93 (4.49%) indicates there was an improvement in the average engagement of students participating in the program post intervention. When analysing students on the autism spectrum data in isolation, no statistically significant differences were found in students self-report school engagement and belonging; with scores for SEI-E and Belonging measures staying the same or reducing slightly (see S6 Table).

A statistically significant improvement in students' responses to the In My Shoes situation-based scale was noted, which indicates an improvement in students understanding of intervention content; selecting responses that demonstrate behaviour that would lead to the inclusion of their peers in various social situations. Students' confidence in asking for help and helping others in the classroom and playground reduced slightly post intervention. No change was reported in students' self-perceived school involvement. A statistically significant improvement in students' understanding of autism was reported following the second lesson of the intervention. Refer to Table 5 for differences in HCSBS, SEI-E, Belonging, and self-developed scales pre-post intervention across the sample.

No statistically significant differences were found between students on the autism spectrum and typically developing peers post intervention across all measures. SEI-E total scores declined for students on the autism spectrum but improved for peers. While differences between students on the autism spectrum and peers SEI-E scores did not reach significance, scores for both samples moved in a positive direction for most subscales. Refer to S7 Table for differences between samples in HCSBS, SEI-E, Belonging and self-developed scales post intervention.

*Changes in social skills and behaviour in the home, community and school environment reported using HCSBS and SSBS.* Parents of students on the autism spectrum and typically developing peers reported an improvement in their child's self-management and compliance and reported less defiant/disruptive and anti-social behaviour on the HCSBS post intervention. No statistically significant changes were reported in the social skills of students on the autism spectrum on the SSBS by teachers post intervention.

## Discussion

The importance of school connectedness for students social, emotional, and academic development is undisputed [7]. Limited school-based interventions exist that specifically aim to increase elementary school students' sense of connection to school [13, 14]. This study focused on evaluating the feasibility, fidelity and preliminary effectiveness of a novel school-based intervention entitled *In My Shoes* that aims to improve the school participation and feelings of connectedness of students on the autism spectrum. Findings from this study are encouraging, suggesting *In My Shoes* is a feasible intervention and shows promise in improving self-report school engagement of all student participants, as well the classroom participation and subjective school experiences of students on the autism spectrum.

**Table 5. Difference in HCSBS, SEI-E, Belonging, In My Shoes scales pre-post intervention across whole sample.**

| Measures | Pre Mean (SD) | Post Mean (SD) | Z score | P value |
|---|---|---|---|---|
| **HCSBS (total n=27; 9 students on autism spectrum and 18 typically developing peers)** | | | | |
| Peer relations | 64.70 (14.09) | 65.37 (13.83) | 1.178 | 0.239 |
| Self-management/ compliance | 54.15 (11.94) | 56.96 (10.84) | 2.361 | 0.018* |
| Social competence total | 118.85 (25.03) | 122.3 (23.66) | 1.750 | 0.080 |
| Defiant/ disruptive | 33.36 (12.38) | 29.96 (10.25) | 2.429 | 0.015* |
| Antisocial/ aggressive | 27.15 (10.26) | 25.74 (8.64) | 1.200 | 0.230 |
| Antisocial behaviour total | 60.52 (21.89) | 55.04 (17.96) | 2.320 | 0.020* |
| **SEI-E (n=200)** | | | | |
| Teacher student relationship | 36.21 (5.91) | 36.55 (6.50) | 0.256 | 0.798 |
| Peer support for learning | 23.62 (4.19) | 24.05 (4.20) | 0.526 | 0.599 |
| Family support for learning | 17.75 (2.39) | 17.70 (2.62) | 0.145 | 0.884 |
| Future goals and aspirations | 20.70 (3.73) | 21.020 (3.43) | 1.143 | 0.253 |
| Intrinsic motivation | 6.74 (3.19) | 8.790 (2.03) | 6.822 | 0.001*** |
| Behavioural engagement | 9.14 (2.36) | 9.23 (2.20) | 0.181 | 0.856 |
| Disaffection | 8.89 (2.96) | 8.95 (2.69) | 0.700 | 0.484 |
| SEI-E total | 105.01 (13.84) | 108.10 (13.62) | 3.317 | 0.001*** |
| Belonging Scale | 30.13 (4.14) | 30.02 (4.61) | 0.289 | 0.773 |
| **In My Shoes (n=200)** | | | | |
| Situation based | 15.65 (2.09) | 16.06 (2.05) | 3.212 | 0.001*** |
| In the past week | 37.01 (5.97) | 36.24 (5.81) | 2.634 | 0.008** |
| Involvement | 20.09 (3.65) | 20.31 (2.89) | 0.235 | 0.814 |
| Learning about the autism spectrum | 7.21 (1.25) | 7.51 (1.16) | 3.492 | 0.001*** |

*Note.*

HCSBS, Home Community Social Behaviour Scale; SEI-E, Student Engagement Instrument – Elementary Version

*p<0.05

**p<0.01

*** p<0.001.

## Feasibility

**Recruitment capability and sample characteristics.** The recruitment process for this study was time consuming and eligibility criteria was restrictive. The primary researcher was required to contact school principals in the first instance to determine interest, eligibility and to gain written informed consent. Due to school principals limited availability, this resulted in several phone calls and emails before the primary researcher was able to communicate with students on the autism spectrum, their parents and teachers and prepare for data collection. The fact that we received several enquiries about our study from schools with students experiencing school participation restrictions, without a formal diagnosis of autism, indicates the need for an intervention that focuses on improving students school participation and feelings of connectedness. Although we were able to recruit sufficient participants for this study, we anticipate challenges recruiting large numbers of schools in future studies without broadening eligibility criteria to include students with social challenges without formal diagnosis of autism. This, however, needs to be considered carefully, as broadening eligibility may impact scientific rigor as we will not be able to differentiate intervention effects for different student populations.

**Data collection procedures and outcome measures.** We received consistent feedback from students and teachers that the timing and quantity of outcome measures were

burdensome. Selecting outcome measures in intervention research is challenging [54]. Several outcome measures are used in feasibility studies to identify the most appropriate measure to use in future efficacy studies [24]. Striking the balance between thorough data collection procedures and feasibility can be particularly challenging in busy school environments. We selected measures that addressed constructs of interest and that had been validated with elementary school students. The measures available, however, had limited psychometric evaluations which may have impacted findings. As we move forward, we may need to develop new measures that align with the theoretical perspectives and hypothesized mechanisms of change reflected in the intervention [23, 54].

Several objective changes were observed in classroom participation of students on the autism spectrum post intervention. Students were observed to display more on task behaviour and interacted more with peers post intervention; a proximal intervention outcome, specifically targeted in the *In My Shoes* intervention. Students also reported via ESM surveys, that when they needed help and classmates helped them more post intervention. This is an important finding, as it suggests peers have an improved ability to demonstrate pro-social behaviour; an intended outcome of the intervention. Social skills, however, such as the ability to adjust to different behavioural expectations explored using the teacher-report SSBS, were not overtly targeted in the intervention and did not change post intervention. The SSBS was recommended in a recent systematic review evaluating the psychometric properties of social skills measures [55]. This distal outcome measure was important to include in this study as it served an important function in determining if intervention effects transcended immediate intervention targets. These findings suggest that for this intervention and sample size, there were no effects in relation to distal outcomes.

Several factors may have contributed to lack of significant change in the self-report school engagement of students on the autism spectrum. For example, students may have misinterpreted survey items or may have experienced difficulty understanding and applying key concepts of the program specifically relating to perspective taking. While interventions that adopt whole class approaches have their advantages, students on the autism spectrum may benefit from additional individualised support throughout the duration of the intervention to specifically apply concepts and practice skills with peers to support change in intervention outcomes over time. This adaptation, however, would need to be tested to evaluate if it is feasible in the school environment.

Some interesting findings arose from ESM data. For example, students reported feeling more interested in classwork, but also reported feeling more worried when participating in classwork post intervention. This may suggest that students care more about their classwork and therefore feel more worried about their performance in the classroom post intervention. Several inconsistencies, however, were noted in data relating to students' emotions from ESM surveys. For example, students reported increased enjoyment when with their classmates', but at the same time reported feeling angrier when with their classmates post intervention. Although students had appropriate reading comprehension and were provided with training in the use of the ESM survey, inconsistencies suggest more training is needed to support students to interpret emotion-specific items.

**Appropriateness, implementation, and practicality.** The components of the intervention that were most valued was the whole class program. Teachers valued detailed lesson plans and interactive pre-prepared resources. The whole school and parent component of the intervention, that were less prescriptive and provided schools with flexibility in the way they were delivered, were less valued and therefore not implemented as stated in the manual. This raises important questions about how to best support learning between the classroom and school and between school and home.

All schools and teachers felt that to make a difference at a school level, the intervention needed to be embedded across the school; tailored to as many grade levels as possible. This

would help to develop a shared set of values within the school about how students should respond to and support each other and equip schools with the tools they need to facilitate these behaviour transactions. These findings are consistent with school connectedness literature that suggest whole school approaches targeting school organisational environments are effective in promoting a sense of belonging [56, 57]. It is not reasonable to expect systemic change if content is only delivered to a small number of students. Future research should aim to expand *In My Shoes* across grade levels and provide additional resources to support schools to implement whole school activity ideas. Additional emphasis should also be placed in pre-intervention professional learning on the importance of whole school and parent involvement so that teachers understand the potential impact this could have on intervention outcomes and therefore be more invested in delivering these intervention components. Identifying school leadership staff who will be accountable for implementing whole school activity ideas from the outset would also help to improve the fidelity of this intervention component.

Parents oscillated between wanting to be provided with information and not wanting to be provided with too much information. COVID-19 social distancing restrictions made parent engagement particularly challenging in this study, highlighting how quickly the disconnect between home and school can occur and the amount of effort required in building relationships and sharing knowledge between school and home. Innovative ways to maximise parent engagement, such as presenting written information in functional formats (e.g., condensing weekly information handouts to present key concepts on an A4 sized fridge magnet) and using videos on school portals to demonstrate student learning, should be incorporated in the future.

## Preliminary effectiveness

Despite the small sample size, statistically significant positive change in intervention outcomes were noted across the sample including improved student self-report engagement, intrinsic motivation and understanding of autism. Students' perception of peer support also improved, but this did not reach significance. These findings are encouraging as they show a positive trend in key constructs (e.g., feelings of acceptance, inclusion and belonging, and perceptions of the quality of teacher and peer relationships and support) that contribute towards students' sense of school connectedness [1, 8]. This indicates the intervention has the potential to buffer the long-term documented implications of reduced school connectedness on student outcomes.

While there were some changes to the classroom participation and subjective experiences of students on the autism spectrum, benefits to peers were significant and exceeded expectations. Statistically significant changes were noted in students' self-report engagement and motivation at school post intervention, which was not found when analysing data of students on the autism spectrum in isolation. Unlike some interventions, *In My Shoes* focuses on making change at an environmental level; using a whole class program to teach peers to recognize and respond when a student may be having difficulty in the classroom and playground. In raising peer's awareness and understanding of autism, we can create a more inclusive and supportive classroom environment that fosters participation. Involving peers in school-based interventions and using a top-down approach, focusing holistically on student participation rather than developing a particular set of skills in isolation, is imperative to effect changes in the school experiences of students on the autism spectrum.

## Future research

While it may appear conducting separate feasibility studies prior to launching a randomised controlled trial (RCT) will prolong the research process, a carefully constructed sequence of preliminary studies will ultimately accelerate the development of more effective school-based

interventions [23]. A number of recommendations for future research can be made based on the current study. Firstly, the existing *In My Shoes* intervention should be adapted based on feedback received from parents, teachers, and students (e.g., simplify worksheets; incorporate more technology into lesson plans; condense parent information handouts; expand content to include more grade levels) and then tested in a larger number of schools. If this shows promising results, an RCT may be suitable to further test the interventions effectiveness [58].

Separate studies could then broaden *In My Shoes* eligibility criteria to include other student populations such as students with social challenges without a formal diagnosis of autism and other neuro-diversities such as ADHD. The intervention would need to be adapted based on literature to ensure the intervention is appropriate for these student populations and tested for feasibility and effectiveness in small samples before larger studies are conducted.

Striking a balance between data collection procedures that are thorough but also feasible should be a priority in future studies by reducing the number of paper-based outcome measures and focusing on capturing changes in proximal rather than distal intervention outcomes over time. Measuring outcomes mid-term to mid-term may also help to reduce burden for teachers in the first and final week of term; minimising the impact contextual factors may have on study findings. Future studies involving ESM should provide more in-depth training; supporting students to practice responding to items relating to emotions using real life examples through role play and provide students with the opportunity to practice using the device a few days before data collection starts. Emotion-specific items should also be adapted to use a dichotomous rather than continuous scale and be context and activity specific, rather than asking students to reflect on their emotions more generally.

## Limitations

Conducting research in schools is complex and multifaceted. There are many factors that impact on the delivery of school-based interventions and the collection of data, which can ultimately impact the success of school-based interventions. This is often why intervention research is not commonly conducted in schools and why there continues to be a paucity of interventions that aim to support student's participation and sense of belonging at school [13, 14].

Limitations of the present study must be acknowledged. Only a small sample of students on the autism spectrum across schools participated, which limited the power of the study and may have caused Type I errors. While intervention effectiveness was evaluated, the focus of this study was to evaluate the feasibility of the intervention in the school environment and therefore a small sample was appropriate to determine if larger scale studies are warranted. HLM was attempted with ESM data; however, the sample size was too small to yield meaningful results. Schools that did participate did so voluntarily and therefore inherently may have had a more positive school culture relating to the inclusion of students with additional needs which may have biased results. Practical issues relating to the mEMA app and the electronic platform should also be considered. Several survey instances were missing due to the mEMA app failing to prompt, students not hearing the prompt and/or forgetting to keep the device on them while at school. Typically developing peers were not included in qualitative interviews as it was not deemed feasible for this study, however, will be considered in future pilot studies or RCTs. Further support and training are required to minimize the impact of technical issues on data collection.

## Conclusions

The feasibility, fidelity, and preliminary effectiveness of a novel school-based intervention entitled *In My Shoes* was evaluated in this study. Teachers valued the whole class component of the

intervention, including its detailed lesson plans and pre-prepared interactive resources. This intervention component was delivered as stated in the manual; however, teachers and schools found the parent and whole school component of the intervention more challenging to implement due to lack of time and resources and COVID-19 restrictions. Study findings provide preliminary evidence to support the effectiveness of the intervention in improving student self-report school engagement, motivation and understanding of autism. The intervention shows promise for students on the autism spectrum, improving peer interactions and teacher responsivity, reducing inattentive behaviours and reported difficulties in the classroom. Useful insights into ways the intervention and the design of future research can be improved are discussed.

## Supporting information

**S1 Table. Intervention fidelity protocol.**
(DOCX)

**S2 Table. Teacher, school leadership and parent responses to anonymous post intervention feedback survey.**
(DOCX)

**S3 Table. Student responses to anonymous paper-based post intervention feedback survey.**
(DOCX)

**S4 Table. Difference in objective ESM data pre-post intervention, autism sample.**
(DOCX)

**S5 Table. Difference in subjective ESM data pre-post intervention, autism sample.**
(DOCX)

**S6 Table. Difference SSBS, SEI-E, belonging scale scores pre-post intervention, autism sample.**
(DOCX)

**S7 Table. Difference between autism and TD scores in HCSBS, SEI-E, Belonging, In My Shoes scales pre-post In My Shoes intervention.**
(DOCX)

## Acknowledgments

We are thankful to the schools, students and parents who participated in this study. We would also like to thank Dr. Cally Kent for her assistance in rating the videos.

## Author Contributions

**Conceptualization:** Amy Hodges, Reinie Cordier, Annette Joosten, Helen Bourke-Taylor.

**Data curation:** Amy Hodges.

**Formal analysis:** Amy Hodges, Yu-Wei Chen.

**Investigation:** Amy Hodges.

**Methodology:** Amy Hodges, Reinie Cordier, Annette Joosten, Yu-Wei Chen.

**Project administration:** Amy Hodges.

**Supervision:** Reinie Cordier, Annette Joosten, Helen Bourke-Taylor.

**Validation:** Amy Hodges.

**Visualization:** Amy Hodges.

**Writing – original draft:** Amy Hodges.

**Writing – review & editing:** Amy Hodges, Reinie Cordier, Annette Joosten, Helen Bourke-Taylor, Yu-Wei Chen.

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
