## [Decision Letter · Decision Letter 0]

1 Dec 2021

PONE-D-21-22094

Evaluating the feasibility, fidelity, and preliminary effectiveness of a school-based intervention to improve the school participation and feelings of connectedness of elementary school students on the autism spectrum.

PLOS ONE

Dear Dr. Cordier,

Thank you for submitting your manuscript to PLOS ONE. After careful consideration, we feel that it has merit but does not fully meet PLOS ONE’s publication criteria as it currently stands. Therefore, we invite you to submit a revised version of the manuscript that addresses the points raised during the review process.

Ain revising the manuscript, I would suggest you take particular note of the reviewer's suggestions about strengthening the literature review and rationale for the study as well as providing additional information and clarity about methods and measures used.

We look forward to receiving your revised manuscript.

Kind regards,

Amanda A. Webster

Academic Editor

PLOS ONE

https://journals.plos.org/plosone/s/file?id=ba62/PLOSOne_formatting_sample_title_authors_affiliations.pdf"

2. Please note that in order to use the direct billing option the corresponding author must be affiliated with the chosen institute. Please either amend your manuscript to change the affiliation or corresponding author, or email us at plosone@plos.org with a request to remove this option.

Reviewers' comments:

Reviewer's Responses to Questions

**Comments to the Author**

1. Is the manuscript technically sound, and do the data support the conclusions?

Reviewer #1: Partly

Reviewer #2: No

Reviewer #3: Yes

2. Has the statistical analysis been performed appropriately and rigorously? 

Reviewer #1: Yes

Reviewer #2: I Don't Know

Reviewer #3: Yes

3. Have the authors made all data underlying the findings in their manuscript fully available?

Reviewer #1: Yes

Reviewer #2: No

Reviewer #3: Yes

4. Is the manuscript presented in an intelligible fashion and written in standard English?

Reviewer #1: Yes

Reviewer #2: No

Reviewer #3: Yes

5. Review Comments to the Author

Reviewer #1: Thank you for the opportunity to review Evaluating the feasibility, fidelity, and preliminary effectiveness of a school-based intervention to improve the school participation and feelings of connectedness of elementary school students on the autism spectrum submitted to the Research Article section of PLOS ONE. My comments are below:

You refer to students with autism spectrum disorder (ASD) a few different ways throughout the manuscript. It would help for clarity to simply refer to them as students with ASD as it’s the most current terminology.

I’m confused about the purpose of the In my Shoes curriculum. It states that it’s an online support for teachers and peers, but reads like a social skill training program for students with ASD. Perhaps offer more explanation of how the teachers or peers use the curriculum so as to not confuse the reader.

I don’t see how a training curriculum for teachers and peers builds connectedness for students with ASD. There’s needs to be a strong rationale for how this curriculum can do this, as well as break down connectedness into its component parts and create a rationale for sections of the curriculum targeting each component.

What was the purpose of the qualitative data collection? Was it to better understand answers on the survey, were only certain participants interviewed based on survey data or were they all interview? This question applies to the teachers, parents, and students with ASD as well. What were the procedures for selection and what was the intent beyond gathering more information. Was it to better understand scores on the assessment? Were participants with specific scores targeted for follow up or did it occur for everyone?

Its concerning that there’s an emphasis on school connectedness in the opening and there is no mention of connectedness in the methods section. Why wasn’t a measure of school connectedness included as a part of the battery of instruments used? If the measure used for school engagement and belonging is the proxy measure for school connectedness that needs to be made clear, but also should be defined in the opening section as the author’s definition of school connectedness.

Reviewer #2: Thank you for the opportunity to review this interesting manuscript. The authors have conducted some very creative research on an important and interesting topic, The notion of a universal classroom intervention to improve the school connectedness of children on the spectrum is commendable. There were however a number of difficulties ( in the rationale method and presentation of the results in particular) detailed below that detracted from the creative potential of this work:

1. The abstract was lacking in important details. It needed a clear rationale statement. The age or grade of the sample needed to be identified. The sample description is inadequate. It is not clear whether the 10 students were students with autism. We subsequently learn that data was also collected from about 200 “typically developing “ children, but this is not mentioned or detailed in the abstract. The “preliminary evidence” that is referred to needs to be given some content.

2. The introduction needed a more thorough conceptual or theoretical understanding of belonging and school connectedness in particular. This conceptual understanding is important to provide the reader with a sense of the intervention being implemented. The authors could also have reviewed the literature more extensively on attempts to promote school connectedness, even if these interventions were not targeted specifically at children on the spectrum. I subsequently note that in the method the authors refer to quite extensive research that they have conducted prior to the development of the intervention. A more extensive review of some of this research needed to part of the rationale of the study.

3. Participant recruitment, characteristics and numbers were inadequately detailed in the participant section . We need to know more about the numbers and sample and how the schools and students were selected and recruitment rates (if possible) etc. A particular source of frustration were the lack of details in this section about the recruitment and characteristics of the “typically developing” students. Initially on reading this method section I wondered why the peers were not sampled as part of the outcome, as they formed such an important part of the rationale? Later there is a suggestion that measures were given to non autistic children but this is not presented in the sample characteristics in the method section. We learn in the results and the supplementary table that 200 student surveys were completed? Who are these students? How were they recruited and why are we not provided with any participant characteristics in this regard? Were they part of the informed consent procedure?

4. Rationale for the screening measures needed to be more clearly articulated eg. The teacher efficacy scale.

5. The questions around “feasibility” seemed more akin to questions around process evaluations and acceptability rather than feasibility - this construct needs to be more clearly defined and operationalised.

6. There was some great creativity in some of the measures used such as ESM technology. However there generally needed to be a clearer link between program rationale and expected outcomes and the measures

7. Many measures had no sample items- it would have helped the reader to get sample items on all measures .

8. The behavioural observation measures were interesting- what steps however were taken by the rater/s to avoid the likelihood of confirmation bias?

9. With regard to the results -the tables generally needed to contain the Ns- whether it be with the teachers, students on the spectrum, the “typically developing” students. As indicated above the sample characteristics of the typically developing sample needed to be detailed. It was difficult to work out at times who the student participants were and their number for each of the measures and analyses. It is only when you dig into the supplementary material that you get a sense of the wider evaluation and comparison with the typically developing students. It is unfortunate that this was not presented more clearly.

10. The discussion was relatively solid although it was at times difficult to determine the accuracy of the interpretation because the sample and results were not well presented.

11. The limitations needed to be expanded upon. In particular the small sample size limitation was understated. There were so many measures with such a small sample that the notion of type 1 error needed to be discussed. Other limitations such as the lack of a comparison group in pre and post measures also needed to be stated more strongly – even if this is a preliminary study. Do thesese pre and post outcome measures reflect any reliable or or meaningful change

Overall this was a potentially creative study that could make an important contribution but needed to be written more tightly and clearly. A revision could possibly benefit from also dropping the pre and post outcome measures on a sample that is so small and without a comparison group.

Reviewer #3: The article presents the results of a feasibility study of the In My Shoes program for primary age students. The paper is very well presented and is a convincing and very detailed description of the study and the initial results of the feasibility study.

However, there is little description of the previous literature or the theoretical underpinning of the In My Shoes program. I believe the paper would be improved if these two areas were addressed. Inclusion of the extant literature would provide a stronger rationale for the development of the program.

As I have already stated the description of the program and the implementation has been described in great detail.

I think that some careful editing of the Procedures and Measures section would be beneficial.

It is obvious that the research was strong but the overwhelming detail in these sections is more appropriate for a thesis rather than a paper.

I congratulate the authors on the high standard or presentation and the relevancy of the program for students with autism in primary schools

6. PLOS authors have the option to publish the peer review history of their article (what does this mean?). If published, this will include your full peer review and any attached files.

Reviewer #1: No

Reviewer #2: No

Reviewer #3: **Yes: **Roselyn May Dixon

---

## [Author Response · Author response to Decision Letter 0]

13 Mar 2022

Thank-you for reviewing our manuscript, ‘Evaluating the feasibility, fidelity, and preliminary effectiveness of a school-based intervention to improve the school participation and feelings of connectedness of elementary school students on the autism spectrum’. We have responded to comments and revised the manuscript accordingly (please find attached response to reviewers document). We have listed reviewer comments and our response and have cross referenced line numbers where the changes were affected in the marked manuscript. Revisions following specific reviewer recommendations are shown using track changes in the attached manuscript.

---

## [Decision Letter · Decision Letter 1]

16 May 2022

Evaluating the feasibility, fidelity, and preliminary effectiveness of a school-based intervention to improve the school participation and feelings of connectedness of elementary school students on the autism spectrum.

PONE-D-21-22094R1

Dear Professor Cordier,

I am pleased to inform you that your manuscript has been judged scientifically suitable for publication and will be formally accepted for publication once it meets all outstanding technical requirements. Although I have only received one review, I feel that the comments are sufficient to indicate you have addressed the previous issues. I also do not feel it is fair to prolong this as I am struggling to find a second reviewer, and have thus decided to accept the recommendations of one reviewer.

Kind regards,

Amanda A. Webster

Academic Editor

PLOS ONE

Additional Editor Comments (optional):

Reviewers' comments:

Reviewer's Responses to Questions

**Comments to the Author**

1. If the authors have adequately addressed your comments raised in a previous round of review and you feel that this manuscript is now acceptable for publication, you may indicate that here to bypass the “Comments to the Author” section, enter your conflict of interest statement in the “Confidential to Editor” section, and submit your "Accept" recommendation.

Reviewer #1: All comments have been addressed

2. Is the manuscript technically sound, and do the data support the conclusions?

Reviewer #1: Yes

3. Has the statistical analysis been performed appropriately and rigorously? 

Reviewer #1: Yes

4. Have the authors made all data underlying the findings in their manuscript fully available?

Reviewer #1: Yes

5. Is the manuscript presented in an intelligible fashion and written in standard English?

Reviewer #1: Yes

6. Review Comments to the Author

Reviewer #1: Thank you for the opportunity to re-review Evaluating the feasibility, fidelity, and preliminary effectiveness of a school-based intervention to improve the school participation and feelings of connectedness of elementary school students on the autism spectrum submitted to the Research Article section of PLOS ONE. The authors have sufficiently addressed my comments and enhanced their manuscript, I have no further recommendations.

7. PLOS authors have the option to publish the peer review history of their article (what does this mean?). If published, this will include your full peer review and any attached files.

Reviewer #1: No

---

## [Editor Report · Acceptance letter]

23 May 2022

PONE-D-21-22094R1 

Evaluating the feasibility, fidelity, and preliminary effectiveness of a school-based intervention to improve the school participation and feelings of connectedness of elementary school students on the autism spectrum. 

Dear Dr. Cordier:

I'm pleased to inform you that your manuscript has been deemed suitable for publication in PLOS ONE. Congratulations! Your manuscript is now with our production department. 

Kind regards, 

on behalf of

Dr. Amanda A. Webster 

Academic Editor

PLOS ONE